# Maternal saliva visfatin level in term and preterm labor: A case control study

**Khadijeh Nasri[1], Mona Mehrabi[2], Mojtaba Bayani[3], Amir Almasi-Hashiani[4,5]***

**1** Department of Obstetrics and Gynecology, Arak University of Medical Sciences, Arak, Iran, **2** Student Research Committee, Arak University of Medical Sciences, Arak, Iran, **3** Department of Periodontics, School of Dentistry, Arak University of Medical Sciences, Arak, Iran, **4** Department of Epidemiology, School of Health, Arak University of Medical Sciences, Arak, Iran, **5** Traditional and Complementary Medicine Research Center, School of Medicine, Arak University of Medical Sciences, Arak, Iran

* Amiralmasi2007@gmail.com

**Data Availability Statement:** All relevant data are within the manuscript and its Supporting Information files.

**Funding:** This study was funded by Vice-chancellor for Research of Arak University of Medical

## Abstract

Visfatin, a colony-enhancing factor (pre-B-cell), is an inflammatory biomarker that is secreted from a different number of cells and appears to have some proinflammatory or immune-regulating effects. The aim of this study was to compare maternal saliva visfatin levels in women with preterm and term delivery. In This case-control study, women in labor before 37 weeks of gestation were the case group (n = 40) and women in labor after 37 weeks of gestation were in control group (n = 40). The saliva of the participants was sampled and maternal saliva visfatin level was measured by ELISA test. In this study, 80 pregnant women were studied in case and control groups. The mean age of case and control group was 29.1±6.9 and 30.55±5.3 years, respectively. The results revealed that the mean maternal saliva visfatin level in the preterm group (4.75±2.2) is significantly (p = 0.001) lower than that in term birth group (6.7±3.1). The results of adjusted logistic regression revealed that after adjusting for GDM, preeclampsia, pre pregnancy BMI and weight gain, the mean of maternal saliva visfatin level in the preterm group is significantly lower than that in the term group (p = 0.026). Considering that during the term pregnancy visfatin levels increase and visfatin may play a role in initiating labor, in our study due to the high visfatin level in case group although the level of maternal saliva visfatin was lower than the control group but high levels of visfatin in the case group can represent the role of visfatin in initiating labor and due to this issue can be use the role of this adipokine for early diagnosis of preterm delivery can be used to prevent, treat and improve the prognosis of this disease. Also, this study is the first study to compare the maternal saliva visfatin level between SGA and AGA group and there is no difference between these groups.

## Introduction

Nicotinamide Phosphoribasyl Transferase (NAmPRtase or NampT) or Pre-B-cell Colony Enhancing Factor 1 (PBEf1) or visfatin is a protein mainly produced in visceral fat tissue. Visfatin is also found in skeletal muscles, liver, bone marrow and lymphocytes. The role of visfatin has been demonstrated in regulating cell growth, apoptosis and angiogenesis in mammalian

Sciences. The funders had no role in study design, data collection and analysis, decision to publish, or preparation of the manuscript. All authors receive a salary from Arak University of Medical Sciences.

**Competing interests:** The authors have declared that no competing interests exist.

cells [1–3]. Visfatin is a multi-functional inflammatory mediator that can play roles such as growth factor, enzyme or cytokine in different situations in the body [4]. Visfatin is related to the occurrence of several inflammatory diseases and also imitates a role similar to insulin [5]. The concentration of visfatin in saliva or serum increases in many inflammatory, infectious or immunodeficiency diseases, including periodontal diseases, diabetes, cardiovascular diseases, etc. [6].

Today, a high percentage of pregnant women experience premature labor, which is called preterm labor, and this problem causes many problems for both the mother and her premature baby such as mortality and morbidity [7]. Premature birth means delivery before the 37th week of pregnancy [8]. Premature birth is one of the main causes of death in infants without abnormalities, and after birth defects, it is the second cause of infant death, which imposes a high economic and psychological cost on the society [9]. Premature birth is an adverse event pregnancy outcome worldwide with a prevalence of 15 million per year [10]. Various factors, including the advanced maternal age [11–13] and infection [14–16], history of still birth, miscarriage, preeclampsia, placenta previa and multiple pregnancy [15] play a role in preterm delivery.

Visfatin plays a role as a regulator of immune responses and infection-related inflammations, and especially visfatin increases in cases of premature delivery caused by amnionitis [17]. It has been reported that the concentration of visfatin in umbilical cord blood is lower in preterm fetuses than in term fetuses, and visfatin in maternal and fetal circulation may play an important role in the occurrence of preterm birth or premature rupture of fetal membranes [18].

The detection of diagnostic biomarkers for the early identification of groups with high-risk preterm birth, to prevent the birth of premature babies, which incur a lot of economic and psychological costs on the society, is of particular importance. Based on this and since there has not been a study to investigate the level of maternal saliva visfatin of women with premature birth, and the role of visfatin as a multi-functional mediator, in this study we aimed to compare the level of maternal saliva visfatin in premature delivery women with women with term delivery.

## Materials and methods

### Study design and setting

This study is a case-control study. In this study, women who were admitted to Taleghani Hospital (Arak, Iran, 2021) due to spontaneous premature delivery were included in the study. Their criteria for spontaneous premature delivery were confirmed by a gynecologist or senior gynecology resident. A detailed examination of teeth and gum diseases was performed by a dentist. The case group includes women who gave birth before 37 weeks of pregnancy and the control group includes women who gave birth after 37 weeks of pregnancy. This study has been approved by the Ethics Committee of Arak University of Medical Science with ID: IR. ARAKMU.REC.1399.346. The authors had access to information that could identify individual participants during the data collection but after that, the data coded and access to information that could identify individual participants was limited to supervisor.

### Participants

These women were included in the study after being approved by an assistant gynecologist by checking the inclusion criteria. The inclusion criteria included the following: women with premature labor and term delivery, having informed consent to participate in the study, natural pregnancy (not assisted reproductive technology), singleton pregnancies, absence of systemic diseases such as metabolic syndrome, diabetes mellitus, autoimmune disorders, acute and

chronic infections, malignancies, heart diseases, liver diseases, etc., not using cigarettes, absence of chorioamnionitis, absence of connective tissue diseases and gum diseases and not using drugs that affect fat metabolism, such as steroids and systemic retinoids. In all participants, labor had started spontaneously and iatrogenic preterm birth were not included in the study. Patients' recruitment was lasted from March 2021 to August 2021.

## Saliva sampling

To saliva sampling, the participants asked to avoid food and water for at least 2 hours before taking the saliva sample. To obtain a saliva sample, we first ask the participants to swish 10 sterile normal saline solutions in their mouth for 60 seconds. Next, check the person's mouth to make sure there are no food or water residues. Then, we ask the participant to suck a 1*1 cm piece of paraffin for 90 seconds and then pour 5 ml of his saliva sample into a sterile, dry polyethylene vial. Finally, the obtained samples are quickly transferred to the freezer or -70˚C temperature.

## Laboratory procedures for visfatin level measurement

The samples obtained from the saliva of the subjects in the study were evaluated using the ELISA method. Currently, the only test that is accurate and scientific, does not need to spend a lot of time and money, and it is easier than other tests due to the lack of complex and expensive devices is an ELISA test. This test is similar to other Radio Immuno Assay tests in terms of working, but instead of radioisotope, the color reaction caused by the effect of the enzyme on the substrate is used as an indicator. The intensity of the color also depends on the concentration of the antigen, and usually an ELISA reader is used to measure the intensity of the color created, and the amount of liquid in each absorbent paper is measured using calibrated device (Periotron TM 6000 Proflow Inc, Amityville, NY, USA).

## Sample size

To determine the required sample size, type one error was considered as 0.05, study power 90%, and based on a study conducted by Pavlová et al. [18], mean maternal visfatin concentration in the group with term delivery was considered as 1.70 ± 0.91 and in preterm birth women 2.83 ± 1.95 ng/ml. The required sample size of each group was 40 woman and a total of 80 people were included.

## Statistical analysis

Mean, standard deviation, count and percentage were used to describe the data. Likelihood ratio Chi-square tests, independent two samples t-test and logistic regression analysis were used to compare the desired variables between the two groups. Statistical tests were performed at a significance level of 0.05 using Stata software version 14 (Stata Corp, College Station, TX, USA).

## Results

In this case-control study, 40 pregnant mothers in each group (preterm and term) were included in the analysis. The comparison of the two groups in terms of demographic and clinical variables is shown in **Table 1**. The analysis did not show any significant difference between the two groups in terms of mean age (p = 0.296), place of residence (p = 0.469), occupation (p = 0.898) and education level (p = 0.968). Regarding the clinical variables, the distribution of endometriosis (p = 0.237), PCO (p = 0.454), history of surgery (p = 0.745), preeclampsia

**Table 1. The comparison of demographic and clinical characteristics among two groups.**

| Variables | | Control Group (n = 40) | Case Group (n = 40) | P value |
|---|---|---|---|---|
| **Age** | **Mean±S.D** | **29.1±6.9** | **30.55±5.3** | **0.296** |
| Place of residence | Urban | 29 (72.5) | 26 (65.0) | 0.469 |
| | Rural | 11 (27.5) | 14 (35.0) | |
| Education | Under diploma | 18 (45.0) | 17 (42.5) | 0.968 |
| | Diploma | 13 (32.5) | 14 (35.0) | |
| | Academic | 9 (22.5) | 9 (22.5) | |
| Mother's job | Housewife | 35 (87.5) | 36 (90.0) | 0.898 |
| | Unemployed | 2 (5.0) | 2 (5.0) | |
| | Employed | 3(7.0) | 2 (5.0) | |
| Gestational age | Mean±S.D | 38.6±0.98 | 32.7±3.1 | 0.001 |
| Endometriosis | Yes | 1 (2.5) | 0 | 0.237 |
| | No | 39 (97.5) | 40 (100.0) | |
| PCO | Yes | 3 (7.5) | 5 (12.5) | 0.454 |
| | No | 37 (92.5) | 35 (87.5) | |
| History of gynecologic surgery | Yes | 5 (12.5) | 6 (15.0) | 0.745 |
| | No | 35 (87.5) | 34 (85.0) | |
| GDM | Yes | 5 (12.5) | 15 (37.5) | 0.009 |
| | No | 35 (87.5) | 25 (62.5) | |
| Pre pregnancy BMI | Under weight | 0 | 5 (12.5) | 0.061 |
| | Normal | 22 (55.0) | 18 (45.0) | |
| | Over Weight | 11 (27.5) | 10 (25.0) | |
| | Obese | 7 (17.5) | 7 (17.5) | |
| Weight gain | Low | 8 (20.0) | 10 (25.0) | 0.024 |
| | Normal | 14 (35.0) | 23 (57.5) | |
| | High | 18 (45.0) | 7 (17.5) | |
| Preeclampsia | Yes | 5 (12.5) | 7 (17.5) | 0.530 |
| | No | 35 (87.5) | 33 (82.5) | |
| Type of delivery | CS | 12 (30.0) | 12 (30.0) | 1.0 |
| | NVD | 28 (70.0) | 28 (70.0) | |

(p = 0.530), pre pregnancy BMI (p = 0.061) and type of delivery (p>0.999) was similar between the two groups. Meanwhile, the prevalence of GDM in the pre-term group was significantly (P = 0.009) higher than the group of term infants (37.5% vs. 12.5%). In addition, the mean weight gain during pregnancy in the term delivery group was significantly higher than the pre-term delivery group, which is due to premature delivery (p = 0.024).

The characteristics of babies were compared between the two groups in **Table 2**. The results showed that the mean height (p = 0.001) and head circumference (p = 0.001) of infants in the term delivery group is significantly higher than the preterm group. Also, in term of gender, the percentage of girl babies in the preterm group was higher than that in term group (p = 0.007).

Maternal saliva visfatin level was compared between two groups. The results revealed that the mean maternal saliva visfatin in the preterm group (4.75±2.2) is significantly (p = 0.001) lower than that in term birth group (6.7±3.1) (**Fig 1**) (mean difference: -1.98, 95%CI: -3.16, -0.79). The results of adjusted logistic regression revealed that after adjusting for GDM, pre-eclampsia, pre pregnancy BMI and weight gain, the mean of maternal saliva visfatin level in the preterm group is significantly lower than that in the term group (p = 0.026). Additional analyzes showed that the mean level of maternal saliva visfatin in women with GDM,

**Table 2. The comparison of infant's characteristics among two groups.**

| Variables | | Control Group (n = 40) | Case Group (n = 40) | P value |
|---|---|---|---|---|
| Gender | Girl | 12 (30.0) | 24 (60.0) | 0.007 |
| | Boy | 28 (70.0) | 16 (40.0) | |
| Height, Mean±S.D | | 51.3±2.0 | 47.1±4.9 | 0.001 |
| Head Circumstance, Mean±S.D | | 35.0 (1.1) | 32.7 (3.0) | 0.001 |
| Weight | AGA | 33 (82.5) | 31 (77.5) | 0.837 |
| | SGA | 5 (12.5) | 6 (15.0) | |
| | LGA | 2 (5.0) | 3 (7.5) | |

preeclampsia and SGA was lower in compared to non-affected women, while this observed difference was not significant (**Table 3**).

## Discussion

This study was designed with the aim of comparing the average level of maternal saliva visfatin in two groups of women with term and preterm birth delivery. The main results of this study suggested that the mean maternal saliva visfatin in the preterm group is significantly lower than that in term birth group. Therefore, it may be possible to use visfatin level as a diagnostic marker for premature birth by conducting more precise studies in various populations in the future, and with timely screening and identification of these high-risk women, the level of mortality rate and its adverse outcomes can be reduced.

Mastorakos et al., reported that during a normal pregnancy, when the mother is not obese or diabetic, fat tissue increases, which is associated with increased insulin resistance. According to this study, normal pregnancy is associated with a high concentration of visfatin in the mother, and the concentration of visfatin in the first trimester is a positive predictor of insulin sensitivity in the second trimester [19]. Also, in a similar study conducted by Mazaki-Tovi et al [20], it was shown that visfatin increases in normal pregnancy and the reason for this is the increase in fat tissue, which increases insulin resistance, and this increase in visfatin levels reduces insulin resistance. In our study, the level of visfatin increased during the semester, which is in line with the results of this study.

Katwa et al [21] in their study presented conflicting reports of visfatin levels in pregnancy. Visfatin levels are high in obese and diabetic women. Visfatin is an adipocytokine that is secreted from fat tissue and helps insulin activity during pregnancy and gestational diabetes. Although the mechanism of action of visfatin is to act like insulin and mimic insulin to increase insulin sensitivity, its role in pregnancy is still unclear. While in our study, there was no relationship between visfatin levels in people with and without GDM and it was not the same as the results of the above study. Szamatowicz et al. [22] reported that visfatin levels are elevated in pregnant women, regardless of the level of glucose tolerance (and regardless of gestational diabetes), which was consistent with the result of our study.

In Fasshauer et al study [23], visfatin levels were clearly higher in women with preeclampsia. While in our study, there was no difference between visfatin levels in people with preeclampsia and those without, and it was not consistent with the results of this study. The findings of the Adali et al study [24] showed that plasma visfatin and leptin levels in preeclamptic patients were higher than those who were normotensive, and visfatin and leptin levels in preeclamptic patients with abnormal Doppler were significantly higher than those with normal Doppler, which indicates that increased levels of leptin and visfatin may play a role in the pathogenesis of preeclampsia, and measuring these adipokines may be useful for

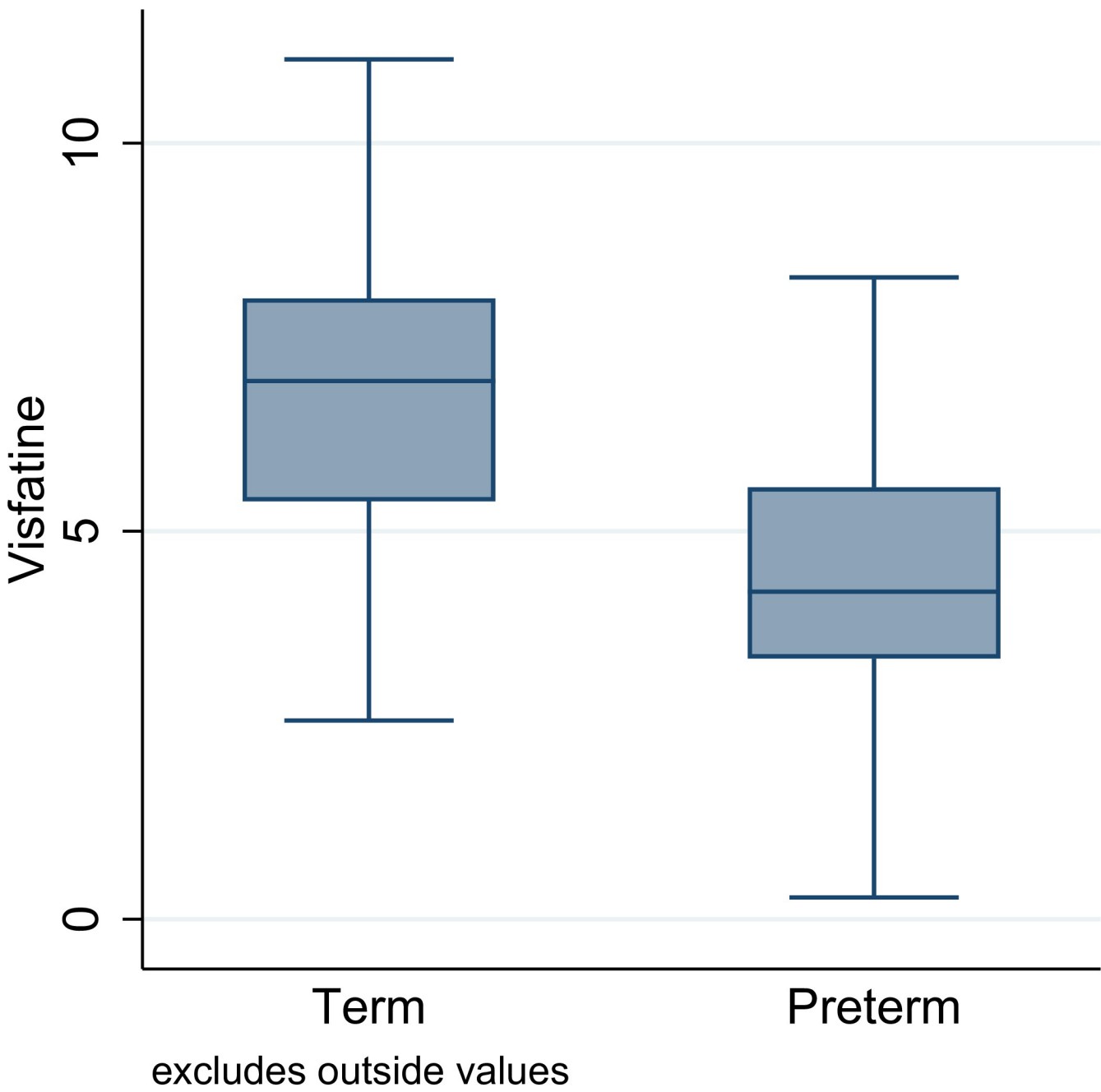

**Fig 1. The comparison of visfatin level based on term and preterm groups.**

measuring the severity of the disease. The results of our study were not consistent with the results of their study.

HU [25] reported that visfatin level was lower in preeclampsia group than both healthy pregnant and non-pregnant groups. In patients with severe pre-eclampsia compared to mild pre-eclampsia, visfatin levels were milder, and according to these findings, there was a decrease in visfatin levels in pre-eclampsia. While in our study, no difference was observed between visfatin levels in women with preeclampsia and those without, and it was not the same as the results of this study.

**Table 3. The comparison of visfatin level based on maternal and infants' outcomes.**

| Outcomes | | Visfatin Level, mean (S.D) | 95% CI for mean | P value |
|---|---|---|---|---|
| Preterm delivery | Preterm | 4.75±2.2 | 4.04–5.45 | 0.001 |
| | Term | 6.73±3.1 | 5.75–7.71 | |
| Weight | AGA | 5.84±2.7 | 5.17–6.52 | 0.316 |
| | SGA | 4.66±3.3 | 2.40–6.91 | |
| | LGA | 6.75±3.2 | 2.74–10.75 | |
| GDM | Yes | 4.82±2.1 | 3.83–5.80 | 0.093 |
| | No | 6.05±3.0 | 5.27–6.82 | |
| Preeclampsia | Yes | 4.42±2.9 | 2.57–6.27 | 0.080 |
| | No | 5.97±2.7 | 5.30–6.64 | |

SD: Standard deviation, CI: Confidence interval

In a study conducted by Malamitsi et al. [26], visfatin levels were significantly higher in pregnancies with intrauterine growth restriction than in pregnancies with AGA babies. Also, visfatin levels in the blood of infants with intrauterine growth restriction on days 1 to 4 are higher than those of AGA infants. Pathological conditions that cause fetal growth restriction can be responsible for increasing maternal visfatin levels. The low level of insulin can be caused by a decrease in the volume of beta cells or dysfunction of beta cells. In our study, there were 11 SGA and 5 LGA cases, and there is no significant difference between the visfatin level of people who had SGA, LGA and AGA and the findings of our study were not consistent with the findings of mentioned study. In the study of Ibáñez et al [27], it was reported that the concentration of visfatin in umbilical cord blood was higher in SGA infants than in AGA infants. While in our study, there was no significant difference between the level of visfatin in the saliva of people who had SGA, LGA and AGA, and it was not consistent with the results of Ibanez study.

In a similar study conducted by Basima et al., the results suggested that the increase in visfatin and CRP levels play a role in the pathogenesis of preterm [28]. Also, in Treza's study in 2015, maternal visfatin level increases during pregnancy and plays an important role in pregnancy complications such as preterm labor, preeclampsia, gestational diabetes, and this marker can be used to detect pregnancy complications in the future. There was no relationship between visfatin level and the presence or absence of preeclampsia, or the presence of gestational diabetes, or the absence of gestational diabetes, which was not consistent with the results of the above study [29].

This study was performed during COVID-19 pandemic. The risk factors associated with adverse fetal outcomes in pregnancies affected by COVID-19 were assessed in previous studies and their findings suggested that gestational age at COVID-19 infection, birth weight, ventilatory supports [30] and symptomatic patients [31] are associated with adverse perinatal outcomes. Also, pregnant women suffering from COVID-19 are more likely to show adverse obstetric and maternal complications [32].

This study had several limitations. One of them was that this study was conducted at the same time as the Covid-19 pandemic, and in some cases, it was difficult to convince pregnant women to participate in the study. In addition, the sample size in this study was calculated in order to compare the mean maternal saliva visfatin level in two term and pre-term groups, and due to the small number of participants with GDM and preeclampsia, as well as the small number of people in weight subgroups (AGA/ SGA/ LGA), the study power in these analyzes is not enough and there is a need to conduct independent studies. One of the strengths of this

study is that limited studies have directly compared visfatin levels in term and pre-term groups, while in this study, the design of the study was based on having term (control group) and pre-term delivery (case group).

## Conclusion

Considering that during the term pregnancy visfatin levels increase and visfatin may play a role in initiating labor, in our study due to the high visfatin level in case group although the level of maternal saliva visfatin was lower than the control group but high levels of visfatin in the case group can represent the role of visfatin in initiating labor and due to this issue can be use the role of this adipokine for early diagnosis of preterm delivery can be used to prevent, treat and improve the prognosis of this disease. Also, this study is the first study to compare the maternal saliva visfatin between SGA and AGA group and there is no difference between this groups.

## Supporting information

**S1 Data.**
(XLSX)

## Acknowledgments

The authors would like to thanks Arak University of Medical Sciences for their scientific support and also we thank the study participants.

## Author Contributions

**Conceptualization:** Khadijeh Nasri, Mona Mehrabi, Mojtaba Bayani, Amir Almasi-Hashiani.

**Data curation:** Khadijeh Nasri, Mona Mehrabi.

**Formal analysis:** Mojtaba Bayani, Amir Almasi-Hashiani.

**Funding acquisition:** Khadijeh Nasri.

**Investigation:** Mona Mehrabi, Mojtaba Bayani, Amir Almasi-Hashiani.

**Methodology:** Khadijeh Nasri, Mona Mehrabi, Mojtaba Bayani, Amir Almasi-Hashiani.

**Project administration:** Khadijeh Nasri, Mona Mehrabi.

**Software:** Mona Mehrabi, Amir Almasi-Hashiani.

**Supervision:** Khadijeh Nasri, Mojtaba Bayani, Amir Almasi-Hashiani.

**Validation:** Khadijeh Nasri, Mona Mehrabi.

**Writing – original draft:** Khadijeh Nasri, Mona Mehrabi, Mojtaba Bayani, Amir Almasi-Hashiani.

**Writing – review & editing:** Khadijeh Nasri, Mona Mehrabi, Mojtaba Bayani, Amir Almasi-Hashiani.

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
