## [Decision Letter · Decision Letter 0]

13 Jun 2023

PONE-D-23-14057Comparison of visfatin levels in saliva in women with term delivery and preterm delivery: a Case-Control StudyPLOS ONE

Dear Dr. Almasi-Hashiani,

Thank you for submitting your manuscript to PLOS ONE. After careful consideration, we feel that it has merit but does not fully meet PLOS ONE’s publication criteria as it currently stands. Therefore, we invite you to submit a revised version of the manuscript that addresses the points raised during the review process.

We look forward to receiving your revised manuscript.

Kind regards,

Antonio Simone Laganà, M.D., Ph.D.

Academic Editor

PLOS ONE

Journal Requirements:

  "This study was funded by Vice-chancellor for Research of Arak University of Medical Sciences."

Additional Editor Comments:

The topic of the manuscript is interesting. Nevertheless, the reviewers raised several concerns: considering this point, I invite authors to perform the required major revisions.

Reviewers' comments:

Reviewer's Responses to Questions

**Comments to the Author**

1. Is the manuscript technically sound, and do the data support the conclusions?

Reviewer #1: Yes

Reviewer #2: Yes

Reviewer #3: Partly

Reviewer #4: Partly

2. Has the statistical analysis been performed appropriately and rigorously? 

Reviewer #1: Yes

Reviewer #2: Yes

Reviewer #3: N/A

Reviewer #4: N/A

3. Have the authors made all data underlying the findings in their manuscript fully available?

Reviewer #1: Yes

Reviewer #2: Yes

Reviewer #3: Yes

Reviewer #4: No

4. Is the manuscript presented in an intelligible fashion and written in standard English?

Reviewer #1: Yes

Reviewer #2: Yes

Reviewer #3: Yes

Reviewer #4: Yes

5. Review Comments to the Author

Reviewer #1: I read with great interest the Manuscript titled " Comparison of visfatin levels in saliva in women with term delivery and preterm delivery: a Case-Control Study" which falls within the aim of the Journal.

In my opinion, this topic analyzed is interesting enough to attract readers’ attention.

Methodology is accurate and conclusions are supported by the data analysis. Nevertheless, authors should clarify some points and improve the quality of the manuscript citing relevant and novel key articles about the topic:

- I suggest a round of language revision, in order to correct few typos and improve readability.

- I would recommend to add further details to discuss common causes of preterm delivery, such as advanced maternal age as well as infections (authors may refer to: PMID: 25027820; PMID: 34207831).

Because of these reasons, the article should be revised and completed. Tables are clear and interesting. Considering all these points, I think it could be of interest for the readers and, in my opinion, it deserves the priority to be published after minor revisions.

Reviewer #2: Dear authors, congratulation

preterm birth is a topic of great interest due to its long term repercussions on the new born

The potential application of the visfatin levels in saliva are of interest and deserve to be published to my opinion

I would like to suggest to rephrase the title

in "maternal saliva visfatin level in term and preterm labor: a case control study"

I would use this rephrase for the entire paper

introduction is nice

methods can you explain how did you calculate the power of the study?

results well presented

conclusion correct

Reviewer #3: I read with great interest the Manuscript titled “Comparison of visfatin levels in saliva in women with term delivery and preterm delivery: a Case-Control Study”, which falls within the aim of this Journal.

In my honest opinion, the topic is interesting enough to attract the readers’ attention. Methodology is accurate and conclusions are supported by the data analysis. Nevertheless, authors should clarify some point and improve the discussion citing relevant and novel key articles about the topic.

Authors should consider the following recommendations:

- Manuscript should be further revised by a native English speaker.

- In the Results section, the Authors have simply reported the p values, from which however it is not possible to deduce the real clinical relevance of the highlighted statistical significance. In order to better understand the obtained results, I suggest reporting not only the p values, but also the corresponding confidence intervals

- The authors have not adequately highlighted the strengths and limitations of their study. I suggest clarifying these points

- Does this manuscript conform the Enhancing the QUAlity and Transparency Of health Research (EQUATOR) network guidelines? It would be mandatory to declare about this element.

-I suggest to discuss, at least briefly, the risk factors associated with adverse fetal outcomes in pregnancies affected by COVID-19 (authors may refer to: PMID: 32975205; PMID: 36143264).

Reviewer #4: The story intends to clarify the mechanism inducing preterm birth from different perspectives. But there are some questions that confused me. First, the authors should explain in detail the inclusion and exclusion criteria, and whether labor was initiated spontaneously. In addition, is the quantity of visfatin different in spontaneous preterm birth and iatrogenic preterm birth? What’s more, the authors should point out the gestational age for sampling. Without it, the authors hardly give a conclusion that the visfatin related to the initiation of labor.

6. PLOS authors have the option to publish the peer review history of their article (what does this mean?). If published, this will include your full peer review and any attached files.

Reviewer #1: **Yes: **Ilaria Cuccu

Reviewer #2: No

Reviewer #3: No

Reviewer #4: No

---

## [Author Response · Author response to Decision Letter 0]

19 Jun 2023

Dear Prof. Antonio Simone Laganà, 

Thank you for giving us the opportunity to submit a revised draft of our manuscript titled “Maternal saliva visfatin level in term and preterm labor: a case control study” to PLOS ONE. 

We appreciate the time and effort that you and the reviewers have dedicated to providing valuable feedback on our manuscript. We are grateful to the reviewers for their insightful comments on our manuscript. Improvements throughout the manuscript were made in accordance with the reviewers’ comments. The comments were extremely valuable for improving the quality of our manuscript. We have been able to incorporate changes to reflect most of the suggestions provided by the reviewers. We have highlighted the changes within the manuscript. 

Here is a point-by-point response to the reviewers’ comments.

Best regards,

Reviewers' comments:

Comments to the Author

1. Is the manuscript technically sound, and do the data support the conclusions?

Reviewer #1: Yes

Reviewer #2: Yes

Reviewer #3: Partly

Reviewer #4: Partly

Response: Thank you for your comment and suggestions that allowed us to greatly improve the quality of the manuscript.________________________________________

2. Has the statistical analysis been performed appropriately and rigorously?

Reviewer #1: Yes

Reviewer #2: Yes

Reviewer #3: N/A

Reviewer #4: N/A

Response: Thank you for your comment and suggestions that allowed us to greatly improve the quality of the manuscript.________________________________________

3. Have the authors made all data underlying the findings in their manuscript fully available?

Reviewer #1: Yes

Reviewer #2: Yes

Reviewer #3: Yes

Reviewer #4: No

Response: Thank you for your comment and suggestions that allowed us to greatly improve the quality of the manuscript.________________________________________

4. Is the manuscript presented in an intelligible fashion and written in standard English?

Reviewer #1: Yes

Reviewer #2: Yes

Reviewer #3: Yes

Reviewer #4: Yes

Response: Thank you for your comment and suggestions that allowed us to greatly improve the quality of the manuscript.

5. Review Comments to the Author

Reviewer #1: 

I read with great interest the Manuscript titled " Comparison of visfatin levels in saliva in women with term delivery and preterm delivery: a Case-Control Study" which falls within the aim of the Journal.

In my opinion, this topic analyzed is interesting enough to attract readers’ attention.

Methodology is accurate and conclusions are supported by the data analysis. Nevertheless, authors should clarify some points and improve the quality of the manuscript citing relevant and novel key articles about the topic:

1- I suggest a round of language revision, in order to correct few typos and improve readability.

Response: Thank you for your comment and suggestions that allowed us to greatly improve the quality of the manuscript. We revised the manuscript accordingly. 

2- I would recommend to add further details to discuss common causes of preterm delivery, such as advanced maternal age as well as infections (authors may refer to: PMID: 25027820; PMID: 34207831).

Response: Thank you for your comment. We added some related explanations to the Introduction in this regard. 

3- Because of these reasons, the article should be revised and completed. Tables are clear and interesting. Considering all these points, I think it could be of interest for the readers and, in my opinion, it deserves the priority to be published after minor revisions.

Response: Thank you for your comment and suggestions that allowed us to greatly improve the quality of the manuscript.

Reviewer #2: 

Dear authors, congratulation

Preterm birth is a topic of great interest due to its long-term repercussions on the new born.

The potential application of the visfatin levels in saliva are of interest and deserve to be published to my opinion.

1- I would like to suggest to rephrase the title in "maternal saliva visfatin level in term and preterm labor: a case control study". I would use this rephrase for the entire paper introduction is nice methods can you explain

Response: Thank you for your comment and suggestions that allowed us to greatly improve the quality of the manuscript. The suggested title was replaced. 

2- How did you calculate the power of the study?

Response: Dear reviewer, thank you for your valuable comments. At the start of study, we calculate the sample size with assuming power of 80% for the main hypothesis. At the end of study, we calculated the power with Stata for the main hypothesis and it was 90.13% as below. 

3- Results well presented.

Response: Thank you for confirming this part of our article.

4- Conclusion correct. 

Response: Thank you for confirming this part of our article.

Reviewer #3: 

I read with great interest the Manuscript titled “Comparison of visfatin levels in saliva in women with term delivery and preterm delivery: a Case-Control Study”, which falls within the aim of this Journal.

In my honest opinion, the topic is interesting enough to attract the readers’ attention. Methodology is accurate and conclusions are supported by the data analysis. Nevertheless, authors should clarify some point and improve the discussion citing relevant and novel key articles about the topic.

Authors should consider the following recommendations:

1- Manuscript should be further revised by a native English speaker.

Response: Thank you for your comment and suggestions that allowed us to greatly improve the quality of the manuscript. We revised the manuscript accordingly. 

2- In the Results section, the Authors have simply reported the p values, from which however it is not possible to deduce the real clinical relevance of the highlighted statistical significance. In order to better understand the obtained results, I suggest reporting not only the p values, but also the corresponding confidence intervals.

Response: Thank you for your comment. We added the 95%CI for the reported mean of visfatin by group, GDM, weight and preeclampsia in Table 3. 

3- The authors have not adequately highlighted the strengths and limitations of their study. I suggest clarifying these points.

Response: Dear reviewer, we added some issues as limitations and strengths to the manuscript (the last paragraph of discussion). 

4- Does this manuscript conform the Enhancing the QUAlity and Transparency Of health Research (EQUATOR) network guidelines? It would be mandatory to declare about this element.

Response: Thank you for your comment. In preparing the manuscript, we followed the STROBE Statement—Checklist of items that should be included in reports of case-control studies and this checklist was submitted to the journal. 

5- I suggest to discuss, at least briefly, the risk factors associated with adverse fetal outcomes in pregnancies affected by COVID-19 (authors may refer to: PMID: 32975205; PMID: 36143264).

Response: Thank you for your comment. We added some related explanations to the discussion section in this regard. 

Reviewer #4: 

The story intends to clarify the mechanism inducing preterm birth from different perspectives. But there are some questions that confused me. 

1- First, the authors should explain in detail the inclusion and exclusion criteria, and whether labor was initiated spontaneously. In addition, is the quantity of visfatin different in spontaneous preterm birth and iatrogenic preterm birth? 

Response: Thank you for your comment and suggestions that allowed us to greatly improve the quality of the manuscript. We clarified the inclusion criteria and it should be highlighted that in all participants, labor had started spontaneously and we did not include iatrogenic preterm birth.

2- What’s more, the authors should point out the gestational age for sampling. Without it, the authors hardly give a conclusion that the visfatin related to the initiation of labor.

Response: Thank you for your valuable comment. In the comparison of visfatin mean between term and pre-term groups, the variables of GDM, preeclampsia, pre pregnancy BMI and weight gain were adjusted by binary logistic regression and gestational age has significant collinearity with groups (term/preterm) and excluded from the model. The mean of gestational age was compared between two groups and as it was reported in Table 1, its mean was 38.6±0.98 and 32.7±3.1 in control and case group.

---

## [Decision Letter · Decision Letter 1]

4 Jul 2023

Maternal saliva visfatin level in term and preterm labor: a case control study

PONE-D-23-14057R1

Dear Dr. Almasi-Hashiani,

We’re pleased to inform you that your manuscript has been judged scientifically suitable for publication and will be formally accepted for publication once it meets all outstanding technical requirements.

Kind regards,

Antonio Simone Laganà, M.D., Ph.D.

Academic Editor

PLOS ONE

Additional Editor Comments (optional):

The authors performed the required corrections, which were positively evaluated by the reviewers. I am pleased to accept this paper for publication.

Reviewers' comments:

Reviewer's Responses to Questions

**Comments to the Author**

1. If the authors have adequately addressed your comments raised in a previous round of review and you feel that this manuscript is now acceptable for publication, you may indicate that here to bypass the “Comments to the Author” section, enter your conflict of interest statement in the “Confidential to Editor” section, and submit your "Accept" recommendation.

Reviewer #1: (No Response)

Reviewer #2: All comments have been addressed

2. Is the manuscript technically sound, and do the data support the conclusions?

Reviewer #1: Yes

Reviewer #2: Yes

3. Has the statistical analysis been performed appropriately and rigorously? 

Reviewer #1: Yes

Reviewer #2: Yes

4. Have the authors made all data underlying the findings in their manuscript fully available?

Reviewer #1: Yes

Reviewer #2: Yes

5. Is the manuscript presented in an intelligible fashion and written in standard English?

Reviewer #1: Yes

Reviewer #2: Yes

6. Review Comments to the Author

Reviewer #1: I read with great interest the Manuscript titled “Maternal saliva visfatin level in term and preterm labor: a case control study”, topic interesting enough to attract readers' attention

The quality of the manuscript has improved thanks to the changes made.

Considering all these points, I think it could be of interest to the readers and, in my opinion, it deserves the priority to be published.

Reviewer #2: Dear authors thank you for addressing reviewers comments. Best regards for your further research.

7. PLOS authors have the option to publish the peer review history of their article (what does this mean?). If published, this will include your full peer review and any attached files.

Reviewer #1: **Yes: **Ilaria Cuccu

Reviewer #2: No

---

## [Editor Report · Acceptance letter]

7 Jul 2023

PONE-D-23-14057R1 

Maternal saliva visfatin level in term and preterm labor: a case control study 

Dear Dr. Almasi-Hashiani:

I'm pleased to inform you that your manuscript has been deemed suitable for publication in PLOS ONE. Congratulations! Your manuscript is now with our production department. 

Kind regards, 

on behalf of

Dr. Antonio Simone Laganà 

Academic Editor

PLOS ONE